# The Combination of Intestinal Alkaline Phosphatase Treatment with Moderate Physical Activity Alleviates the Severity of Experimental Colitis in Obese Mice via Modulation of Gut Microbiota, Attenuation of Proinflammatory Cytokines, Oxidative Stress Biomarkers and DNA Oxidative Damage in Colonic Mucosa

**DOI:** 10.3390/ijms23062964

**Published:** 2022-03-09

**Authors:** Dagmara Wojcik-Grzybek, Magdalena Hubalewska-Mazgaj, Marcin Surmiak, Zbigniew Sliwowski, Anna Dobrut, Agata Mlodzinska, Adrianna Wojcik, Slawomir Kwiecien, Marcin Magierowski, Agnieszka Mazur-Bialy, Jan Bilski, Tomasz Brzozowski

**Affiliations:** 1Department of Physiology, Faculty of Medicine, Jagiellonian University Medical College, 31-531 Cracow, Poland; magdalena.hubalewska@uj.edu.pl (M.H.-M.); marcin.surmiak@uj.edu.pl (M.S.); zbigniew.sliwowski@uj.edu.pl (Z.S.); adrianna.wojcik@uj.edu.pl (A.W.); slawomir.kwiecien@uj.edu.pl (S.K.); m.magierowski@uj.edu.pl (M.M.); 2Department of Microbiology, Faculty of Medicine, Jagiellonian University Medical College, 31-531 Cracow, Poland; anna.dobrut@uj.edu.pl; 3Bioidea Company, 02-991 Warsaw, Poland; agata.mlodzinska@bioidea.com.pl; 4Department of Biomechanics and Kinesiology, Chair of Biomedical Sciences, Faculty of Health Sciences, Jagiellonian University Medical College, 31-008 Cracow, Poland; agnieszka.mazur@uj.edu.pl (A.M.-B.); jan.bilski@uj.edu.pl (J.B.)

**Keywords:** intestinal alkaline phosphatase, voluntary exercise, experimental colitis, diet-induced obesity, inflammation, oxidative stress

## Abstract

Inflammatory bowel diseases (IBD) are commonly considered as Crohn’s disease and ulcerative colitis, but the possibility that the alterations in gut microbiota and oxidative stress may affect the course of experimental colitis in obese physically exercising mice treated with the intestinal alkaline phosphatase (IAP) has been little elucidated. Mice fed a high-fat-diet (HFD) or normal diet (ND) for 14 weeks were randomly assigned to exercise on spinning wheels (SW) for 7 weeks and treated with IAP followed by intrarectal administration of TNBS. The disease activity index (DAI), grip muscle strength test, oxidative stress biomarkers (MDA, SOD, GSH), DNA damage (8-OHdG), the plasma levels of cytokines IL-2, IL-6, IL-10, IL-12p70, IL-17a, TNF-α, MCP-1 and leptin were assessed, and the stool composition of the intestinal microbiota was determined by next generation sequencing (NGS). The TNBS-induced colitis was worsened in obese sedentary mice as manifested by severe colonic damage, an increase in DAI, oxidative stress biomarkers, DNA damage and decreased muscle strength. The longer running distance and weight loss was observed in mice given IAP or subjected to IAP + SW compared to sedentary ones. Less heterogeneous microbial composition was noticed in sedentary obese colitis mice and this effect disappeared in IAP + SW mice. Absence of *Alistipes*, lower proportion of *Turicibacter*, *Proteobacteria* and *Faecalibacterium*, an increase in *Firmicutes* and *Clostridium*, a decrease in oxidative stress biomarkers, 8-OHdG content and proinflammatory cytokines were observed in IAP + SW mice. IAP supplementation in combination with moderate physical activity attenuates the severity of murine colitis complicated by obesity through a mechanism involving the downregulation of the intestinal cytokine/chemokine network and oxidative stress, the modulation of the gut microbiota and an improvement of muscle strength.

## 1. Introduction

Inflammatory bowel diseases (IBD) are a heterogeneous group of disorders of the digestive tract, encompassing two major phenotypic forms: Crohn’s disease (CD) and ulcerative colitis (UC) [1]. Both forms are characterized by a cyclical nature alternating between active and quiescent states [2]. Even though most of IBD’s patients are underweight, the ratio of intraabdominal fat is greater than in healthy individuals [3]. Visceral fat constitutes the major source of pro-inflammatory cytokines [4]. Although the etiology of IBD is still unknown, several factors such as changes in intestinal microbiome, cytokines and oxidative stress were implicated in the pathogenesis of these disorders [5,6,7,8,9,10,11,12,13,14,15,16]. Among therapeutic options, exercise was proposed as one of the most important lifestyle practices to adopt as treatment options for IBD [7,17,18,19,20]. The mechanism of moderate physical activity in amelioration of experimental colitis has been studied by our group before, revealing that voluntary exercise reduced the inflammatory changes in the colon of mice with TNBS colitis [20]. Known for its anti-inflammatory and multi-organ metabolic effects [21], exercise potential benefits for patients with IBD extend beyond metabolic enhancement and include improvements in bone mineral density, fatigue levels and quality of life. There is evidence that short-term combined aerobic and resistance training achieves beneficial changes in body composition in patients with IBD. This is considered as an inexpensive strategy for the prevention and treatment of IBD-related sarcopenia and obesity-related metabolic disorders [7,22]. On the other hand, we have documented that intensive and prolonged exercise escalated inflammatory responses manifested by the increase in the lipid peroxidation and release of inflammatory markers as well as decrease of colonic blood flow (CBF) [20]. As an effect, forced exercise led to exacerbation of colitis and delayed healing of these inflammatory changes in colonic mucosa of obese and non-obese animals. Obesity itself most often related to mesenteric fat in the abdominal cavity seems to contribute to the increased severity of experimental colitis [3]. Voluntary exercise improved colitis in obese mice via diminishing the severity of colonic damage due to an increase in the CBF, at least in part mediated by release of protective myokines [20]. Alkaline phosphatases are enzymes catalyzing the breakdown of monophosphate esters by a hydrolytic removal of phosphate [23]. IAP is known to dephosphorylate toxic microbial cytotoxin, especially lipopolysaccharides (LPS) released from cells walls during stressful events and extracellular nucleotides tri- and diphosphates. It also helps intestinal epithelial cells to create a barrier preventing translocation of bacteria and microbial molecules since this enzyme is enriched in luminal vesicles that are released from the tips of intestinal epithelial cells [24]. Moreover, IAP act as a major regulator of gut intestinal permeability, having impact on the intestinal tight junction [25]. Thus, this enzyme is considered one of the most important factors in the protection of the intestinal mucosa, being responsible for the maintenance of intestinal homeostasis [26]. A decreased expression of IAP is found in disorders such as IBD, metabolic syndrome, cystic fibrosis, necrotizing enterocolitis and diabetes [27,28,29,30,31,32]. Moreover, IAP deficiency has been linked with obesity and IBD [30,33,34]. IAP has also been considered in the mechanism of normalization of intestinal microbiota, thereby regulating the intestinal homeostasis. Interestingly, the IAP-KO mice are characterized by dysbiosis but the administration of exogenous IAP reversed this dysbiosis when compared to wild type mice [35]. Recently, we have reported that voluntary exercise and IAP exerted a beneficial protective effect against inflammatory intestinal changes [36]; however, the alterations in microbiota at the phylum and genus levels, oxidative stress biomarkers, protein expression of proinflammatory factors as well as skeletal muscle strength changes in obesity-induced exacerbation of colitis have not been studied before. Therefore, our present study was designed to evaluate the effect of IAP administration, combined or not with moderate physical activity, on experimental colitis, with a major focus addressed to changes in intestinal microbiota, since microbiota alterations have been associated with the development of intestinal inflammation in experimental murine models of colitis [37]. Moreover, we have examined the changes in the macroscopic and microscopic appearance of colonic mucosa, oxidative stress, proinflammatory cytokines and muscle tension assessed by the grip strength in this animal model of colitis, especially in obese mice subjected to moderate exercise with or without IAP treatment.

## 2. Results

### 2.1. Effects of IAP Administration with or without SW Exercise on Running Distance, Body Weight and Relative Grip Strength in HFD Fed Mice with Colitis 

The effect of voluntary exercise combined with IAP administration on the running distance in mice fed a HFD with or without TNBS-induced colitis is presented in Figure 1. In groups of mice fed an ND, the differences between interventions, including the SW combined or not with IAP treatment, failed to show prominent statistical differences in functional parameters including the running distance, oxidative stress biomarkers and proinflammatory cytokines, and therefore, the only data obtained in HFD fed mice with and without TNBS colitis were included in this presentation. As shown in Figure 1, the group of TNBS colitis mice exercising mice with access to SW expressed a running distance significantly shorter than that recorded in other groups (*p* < 0.05, Figure 1). In obese TNBS colitis mice subjected to SW and IAP administration, a significant increase in the running distance was observed comparing to this distance recorded in mice exercising on SW alone without IAP administration (*p* < 0.05, Figure 1).

Table 1 illustrates the changes in mice body weight monitored in the sedentary or exercising mice with or without IAP administration and TNBS colitis, fed an ND or a HFD at day 5 upon the TNBS or placebo administration. The body weight in sedentary obese mice was significantly increased as compared to the sedentary animals fed a ND. Voluntary exercise on SW alone or the administration of IAP alone or the combination of SW and IAP significantly reduced the body weight in HFD fed mice (*p* < 0.05, Table 1). In contrast, a significant increase in weight was observed in HFD fed mice with colitis treated with IAP as compared to the respective values of weight recorded in TNBS colitis mice fed a HFD or those with TNBS colitis exercising on SW (*p* < 0.05, Table 1).

Figure 2 illustrates the results of the relative grip strength in the normal and obese mice with or without access to SW, the administration of IAP applied alone or in combination with SW (IAP + SW), with or without TNBS colitis. The ND fed mice demonstrated twice as high values for the relative grip strength compared to HFD mice. For all groups exercising on SW with or without colitis, this muscle strength reached higher values compared to groups without SW. The highest relative grip strength was observed in obese mice subjected to a combination of SW and IAP without colitis while the lowest relative values was recorded in sedentary (SEDEN) obese mice with colitis (Figure 2). This relative value of muscle strength in IAP treated obese mice without colitis reached a similar value as recorded in SW obese mice without colitis, while in mice with TNBS-induced colitis, the IAP treatment only slightly tended to increase the value for muscle strength over that measured in the group of HFD fed mice with colitis.

### 2.2. The Effect of Voluntary Exercise Combined with IAP Administration on DAI Activity and Macroscopic and Microscopic Appearance of Colonic Mucosa in Mice with Colitis

Figure 3 shows alterations in the DAI recorded in sedentary and exercising mice, with TNBS-induced colitis, fed a HFD with or without IAP administration or subjected to combination of SW and IAP administration. The DAI score was significantly decreased (*p* < 0.05) in exercising mice and IAP administered alone tended to cause the fall in DAI score below that observed in SW animals, though this change failed to reach a statistical significance. The lowest DAI score was achieved in exercising mice administered with IAP treatment (*p* < 0.05, Figure 3). The DAI score was also significantly decreased (*p* < 0.05) in ND fed mice exercising on SW but the combination of SW and IAP administration failed to significantly affect this parameter as compared with mice exercising on SW only (Appendix A).

Figure 4A–F presents the representative gross macroscopic and microscopic appearance of the colon and colonic mucosa from a sedentary intact mouse fed ND without colitis (A); sedentary control mouse fed a HFD without colitis (B); a mouse fed a HFD with TNBS-induced colitis (C); a mouse fed a HFD subjected to SW with TNBS-induced colitis (D); a mouse fed a HFD administered with IAP with TNBS-induced colitis; and (E) a mouse fed a HFD subjected to the combination of SW and IAP administration with TNBS-induced colitis (F). The colon from the sedentary mouse fed a ND (Figure 4A) as well as from the sedentary mouse fed HFD (Figure 4B) was normal and its mucosa showed a normal appearance without signs of damage. In contrast, the shortening of the colon, with a presence of bloody infusion and bleedings and narrowing of the lumen of the colon, all were observed in mice with TNBS-induced colitis fed a HFD (Figure 4C vs. Figure 4B). In SW exercising obese mice with TNBS colitis, the gross and histologic intestinal injury was reduced (Figure 4C vs. Figure 4D). When IAP alone was administered to mice with colitis, a less bleeding enema and narrower intestinal lumen of the colon, and only minor morphologic mucosal damage, have been observed (Figure 4E). Yet, the clearly visible reduction in the inflammatory reaction, gross lesions and bloody effusion followed by partial reconstruction of colonic mucosa were noticed in obese mice subjected to the combination of voluntary exercise on SW and IAP treatment as compared to those with TNBS colitis (Figure 4C) or exercising on SW only or administered with IAP alone with colitis developed by TNBS administration (Figure 4F vs. Figure 4D and Figure 5E). By histological assessment, the mucosal injury characterized by necrosis of the epithelium and focal lesions of the colonic mucosa with neutrophil infiltration and a loss in the mucosal architecture were observed in obese mouse with TNBS colitis (Figure 4C) compared to mice feed either ND or HFD. These microscopic alternations in the colonic mucosa were attenuated in mice with access to SW (Figure 4D) or after IAP administration (Figure 4E), with less severe damage, moderate neutrophil infiltration and distinguished signs for mucosal regeneration. The minor desquamation of colonic mucosa with regeneration of colonic mucosa and complete lack of bleedings were observed in mice with TNBS-induced colitis subjected to the combination of exercise on SW and IAP treatment (Figure 4F).

### 2.3. Changes in Gut Microbiome in Mice with TNBS Colitis with or without Voluntary Physical Activity on SW, IAP Administration Alone or the Combination of SW and IAP

We investigated whether diet (ND and HFD), voluntary physical activity on SW and the combination of SW and IAP (SW + IAP) administration, may have an impact on microbiome in mice with and without colitis considered changes in microbiota at the phylum and genus levels. We also examined the influence of IAP administration alone on intestinal bacterial composition in selected groups of animals. The analysis of the results obtained from the next generation sequencing (NGS) at the phylum level, where that relative bacterial abundance was examined across all taxa between groups, showed dominance of *Firmicutes* and *Bacteroides* in all studied groups with frequency ranging from 76% in ND group to 95% in the obese TNBS colitis mice exercising on SW (HFD + SW + TNBS) (Figure 5). In turn, *Proteobacteria* were the most common for control ND group and constituted 21% of microbiome composition followed by 15% in mice with colitis fed HFD and 12% in SW exercising obese mice. In other groups, the presence of *Proteobacteria* did not exceed 8%, while *Actinobacteria* occurred in a small percentage in each group (up to 2%) (Figure 5).

We have also investigated microbiome composition at the genera level for all tested groups of mice fed a HFD, only comparing this composition with mice fed a ND without any intervention throughout the experiment (Figure 6). In all groups, dominance of *Bacteroides* and *Clostridium* was noticed. Our analyses showed that the greatest bacterial diversity was found in the ND mice group and distribution of the most numerous genus, which included *Bacteroides*, *Alistipes*, *Ruminococcus*, *Clostridium*, *Faecalibacterium* and *Helicobacter*, was comparable and ranged from 10% to 17%. In other groups, a significant growth of percentage of *Bacteroides* (*p* = 0.000) was observed, whereas the highest, about 5-fold increase in growth was observed in the HFD + SW group in comparison with the ND group. The only group in which *Alistipes* was not detected was in the group of mice with HFD + TNBS colitis; in turn in this group, *Parabacteroides*, *Lactobacillus* and *Desulfiovibrio* were more numerous than in the other groups determined. Moreover, a comparable percentage of *Parasutterella* in ND and HFD + TNBS was observed (Figure 6). Significant differences among studied groups were noticed for *Bacteroides* (*p* = 0.0000), *Lactobacillus* (*p* = 0.0001), *Alistipes* (*p* = 0.0002) and *Parasutterella* (*p* = 0.004).

As shown in Figure 7A,B, the percentage of *Bacteroides* (*p* = 0.032) and pathogenic *Clostridium* genus was increased in obese mice; in turn, the anti-inflammatory *Faecalibacterium* decreased in the HFD group. Analysis of microbiome composition in *Proteobacteria* phylum revealed a simultaneous decrease of *Helicobacter* and lack of *Parasutterella* (*p* = 0.001) in mice fed HFD. The microbiota assessed in groups HFD + SW (Figure 7C) and HFD + IAP (Figure 7D) compared with group HFD + TNBS (Figure 7F) revealed that the most pronounced change was noticed for the *Bacteroides.* In contrast, the abundance of *Alistipes* was the most prominent in the HFD + IAP group, whereas it barely appeared in obese colitis mice. A similar pattern was observed for *Clostridium*, but, in addition, a significant decrease in this genus was observed in exercising animals (HFD + SW) when compared to HFD fed mice (*p* = 0.027). Interestingly, the abundant appearance of *Parasutterella* and an increase in genus *Lactobacillus* and genus *Ruminococcus* mice was observed in TNBS mice as compared to those in SW or IAP-treated obese mice. The gut microbiome composition was comparable in groups of obese mice subjected to the combination SW and IAP (SW + IAP) without TNBS colitis (Figure 7E) or those with TNBS-induced colitis subjected to SW alone (Figure 7G) or the combination of SW + IAP (Figure 7I). Additionally, we noticed a lack of *Parabacteroides* in HFD + SW + IAP with a small detection of this genus in the HFD + SW + IAP + TNBS group. In exercising mice with or without IAP treatment (SW alone or SW + IAP), a significant difference of *Bacteroides* (*p* = 0.003) and *Clostridium* (*p* = 0.02) in comparison with obese mice has been observed. Interestingly, the mice fed ND were the only group showing the presence of *Lactococcus*. *Parabacteroides, Ruminococcus, Desulfovibrio* and *Parasutterella* were not found in stool samples collected from obese mice administered with IAP. Furthermore, *Turicibacter* was the only genus that appeared exclusively in the ND group, but *Sporosarcina* occurred only in ND and HFD fed mice, while *Parvibacter* was detected only in obese colitis mice with or without IAP treatment. Detailed results of a gut microbiome composition in the groups examined in our study is presented on the heat map (Figure 8).

### 2.4. Effects of Voluntary Exercise Combined with IAP Administration on the Mucosal Colonic Content of MDA plus 4-HNE, GSH, SOD Activity and 8-OHdG Concentration in Mice with Experimental Colitis

Figure 9 shows concentration of MDA + 4-HNE in colonic mucosa of mice fed a ND or a HFD alone or without colitis in exercising mice on SW, with or without IAP administration. The content of lipid peroxidation products is significantly higher in the HFD group than in ND mice. Application of voluntary exercise or IAP diminished the level MDA + 4-HNE content in groups of mice with TNBS colitis. The combination of IAP and SW significantly decreased the concentration of MDA + 4-HNE compared to groups with IAP alone or SW alone (*p* < 0.05, Figure 9). The MDA + 4-HNE concentration tended to increase in TNBS sedentary mice fed ND but this increase failed to reach statistical significance compared with mice fed ND without colitis (Appendix A). The concentration of these lipid peroxidation products in mice fed ND was also not significantly affected by administration of IAP alone. However, this MDA + 4-HNE concentration was significantly reduced (*p* < 0.05) in exercising mice and in those subjected to the combination of SW and IAP treatment (*p* < 0.05) (Appendix A).

Total glutathione (GSH + GSSG) concentration in colonic mucosa of mice fed ND or HFD with or without SW or IAP and their combination in mice with TNBS colitis is shown in Figure 10. Group of obese TNBS mice treated with IAP showed significantly higher level of total glutathione compared to the group with colitis without IAP treatment (*p* < 0.05, Figure 10). In TNBS sedentary mice fed ND, the GSH + GSSG concentration tended to increase over that in mice fed ND without colitis, but this increase failed to reach statistical significance (Appendix A). The concentration of GSH + GSSG in mice fed ND was also not significantly affected by either SW and IAP treatment alone or the combination of SW and IAP treatment (Appendix A).

The data on SOD activity in colonic mucosa of mice fed a ND or a HFD alone and obese TNBS mice with access to SW, with or without IAP administration, are shown in Figure 11. An insignificant increase in activity of SOD was observed in obese TNBS colitis mice as compared with the respective group of HFD fed mice without colitis (Figure 11). A group of obese TNBS mice with access to SW administrated with IAP significantly decreased the SOD activity in colonic mucosa (*p* < 0.05, Figure 11). In TNBS sedentary mice fed ND, the SOD activity only tended to increase insignificantly compared with mice fed ND without colitis (Appendix A). The SOD activity in TNBS colitis mice fed ND was significantly reduced (*p* < 0.05) by the exercise on SW with and without the administration of IAP; however, the combination of SW and IAP in these mice was not significantly better in respect to SOD activity than SW or IAP applied alone (Appendix A).

Figure 12 shows DNA oxidative damage expressed as concentration of 8-OHdG in colonic mucosa of sedentary ND or HFD mice TNBS mice running on SW or treated with IAP alone or subjected to the combination of SW and IAP treatment (SW + IAP). Obese mice revealed significantly higher content of 8-OHdG compared to ND mice (*p* < 0.05, Figure 12). Groups of TNBS mice with access to SW or those with IAP administration alone indicated significantly lower DNA oxidative damage compared to the group of obese mice with TNBS colitis alone or those subjected to SW only (*p* < 0.05, Figure 12). The combination of SW and IAP further significantly decreased the concentration of 8-OHdG reflecting reduced DNA damage as compared to 8-OHdG content obtained mice subjected to SW only (*p* < 0.05, Figure 12).

### 2.5. Concentration of IL-2, IL-6, IL-10, IL-12p70, IL-17a, TNF-α, MCP-1 and Leptin in Plasma in Mice with Experimental Colitis, Exercising on SW and Administered with IAP Alone or in Combination with SW

The results presented in Figure 13 show the plasma level of IL-2, IL-6, IL-10, IL-12p70, IL-17a, TNF-α, MCP-1 and leptin in mice fed an ND or a HFD alone or with experimental colitis, exercising on SW with or without administration of IAP. Concentration of IL-2 is significantly lower for the HFD group compared to the ND group (*p* < 0.05, Figure 13). IL-6 concentration is the lowest in both group of animals fed a HFD with colitis and IAP administration and it is significantly lower than in the group with only access to SW (*p* < 0.05, Figure 13). The plasma level of IL-10 significantly increased in the HFD +TNBS group compared to the HFD group but significantly decreased in groups with SW and IAP, with the lowest concentration in the group with both interventions. IL-12p70 concentration showed statistically significant differences in both groups with access to SW or IAP administration compared to group HFD + TNBS, with the lowest level in the group with both factors (*p* < 0.05, Figure 13). IL-17a demonstrates a higher level in the group of mice fed a HFD and colitis compared to the group only fed a HFD and it is significantly lower in the group with IAP compared to the group HFD + TNBS. For the group with both access to SW and IAP administration, the concentration of IL-17a is significantly lower than in the groups with only SW or only IAP (*p* < 0.05, Figure 13). Concentration of TNF-α differs only for the group of sedentary, obese animals compared to sedentary animals fed ND. In both groups with IAP administration, mean concentration is lower, but this difference is not statistically significant. The plasma level of the MCP-1 level is significantly higher in the colitis mice group fed a HFD (*p* < 0.05). A lower mean of concentration appears in the group of IAP administered colitis mice fed a HFD, and this difference reached statistical significance (*p* < 0.05, Figure 13). Leptin level significantly increased in the plasma of obese mice with colitis compared to obese sedentary mice. In the group HFD + IAP + TNBS, it is significantly lower compared to the group HFD + TNBS. For both groups with IAP administration, the leptin mean is significantly lower compared to the HFD + SW + TNBS group (*p* < 0.05, Figure 13).

## 3. Discussion

Our present study demonstrates that experimental colitis in obese mice was ameliorated by the combined application of moderate physical activity on spinning wheels with IAP and that this effect was accompanied with changes in intestinal microbiota, a profound reduction of plasma levels of proinflammatory biomarkers and the attenuation of oxidative stress markers in the colonic mucosa. 

Commencing the animals’ general condition, colitis in mice with only access to SW showed significantly shorter running distance than other groups but this distance returned to comparable values in mice administered with IAP. Furthermore, the reduced running distance was paralleled by a decrease in the muscle strength assessed by the grip test in these mice compared with healthy mice. The running distance on the SW of healthy mice well corresponded with the similar range, 1.3 and 1.1 km/day, reported before in mice without colitis [38]. The access of mice to SW in all groups led to a reduction of their body weights, especially notable in those fed with diet-induced obesity. The sedentary obese mice with colitis presented as much weight loss in 5 days as exercise for 7 weeks in the group HFD + SW, confirming the severity of this model reported previously [39]. Importantly, the IAP administered alone in obese colitis animals prevented the loss of body weight. In fact, treatment with this enzyme has been shown to counteract adipose tissue accumulation, and reduce both insulin resistance and the development of metabolic syndrome. Moreover, IAP knockout mice have been reported to gain weight when fed HFD [27]. The fact that our sedentary mice have been maintained on HFD to the end of the experiment can be explanatory for a gain of the weight of IAP-treated mice over the duration of our study. In contrast, weight loss was not observed in exercising animals with colitis when IAP was administered alone or combined with voluntary exercise on SW, indicating the additional beneficial effect of these two factors in maintaining body weight in experimental colitis. Obese sedentary animals with colitis had decreased the skeletal muscle strength evaluated by the relative grip strength. Indeed, the reduced dynamometric grip force in clinical pediatric patients and, in particular, for male CD patients, during remission or with only mild disease activity, has been reported before [40], reflecting the functional deficits of the skeletal muscle mass in IBD. In our study, the relative grip strength in exercising mice with IAP treatment was greater than the values observed in non-exercising groups, indicating a better skeletal muscle efficiency and the improvement of the general well-being of these animals. 

We have confirmed our previous observation [36] that the combination of voluntary physical activity, along with IAP administration, improved DAI as compared with mice accessing SW only, or those administered with IAP alone against colitis, as reflected by gross macroscopic and microscopic assessments. This marked improvement in colitis was manifested by a reduction in mucosal bleeding and hemorrhagic lesions, less inflammatory reaction, white blood cells’ mucosal infiltration and greater signs of regeneration of damaged colonic mucosa. In colitis mice, the colonic mucosa was damaged and distorted, revealing crypts damage, longitudinal streaky ulcers turning into cavities, erosions and a series of inflammatory infiltrates caused by activated neutrophils: all these features characteristic of inflammatory changes that mimic the course of IBD in humans [41]. Exercise alone ameliorated these inflammatory changes, confirming our previous observation that voluntary physical activity can attenuate the severity of colonic damage in mice fed a HFD [20]. Presumably, this intestinal protection can be mediated by a release of myokines such as irisin from working out skeletal muscle in mice undergoing a moderate voluntary effort on running wheels [20]. Results obtained in the present study seem to indicate that the combination of moderate physical activity on SW with IAP treatment is even more effective in the mechanism of resolution of murine colitis. Indeed, IAP has been reported to afford a protective effect against local and systemic inflammation in systemic infections and various disorders, mainly due to detoxification of bacterial LPS, dephosphorylation of proinflammatory nucleotides, regulation of bicarbonate secretion and duodenal surface pH, absorption of intestinal long-chain fatty acids and regulation of gut microbiome [26]. Recent evidence indicates that this enzyme anti-inflammatory activity against LPS may involve autophagy [42]. In IBDs, the increased levels of extracellular nucleotides such as ATP and UDP are directly involved in protection against intestinal inflammation [43]. The release to these nucleotides is enhanced by intestinal damage and cellular death associated with IBD, in a condition where they were shown to bind receptors on macrophages, epithelial cells and infiltrating T-cells. The IAP has been shown to counteract the subsequent inflammatory responses induced by these nucleotides [43]. Our present findings are corroborative of previous observations that the treatment with exogenous IAP can improve inflammation and strengthen the gut barrier function and integrity in murine colitis, but mechanisms of how IAP can be translated to the human IBD scenario remain unknown [31,44]. Our present study clearly documented that this protective effect of IAP combined with moderate exercise is associated with a fall in the plasma levels of proinflammatory biomarkers and prominent attenuation of oxidative stress markers. 

The physical barrier of the colon is maintained by a single layer of intestinal epithelial cells together with mucus, separating the microbiota from the mucosa, and excluding commensal microbes from penetrating host tissue. Moreover, it integrates incoming signals from commensals, pathogens and dietary components. Sensing of microorganisms plays an important role in cytokine production responsiveness by immune and intestinal cells. Abovementioned processes are strictly regulated by cytokines and growth factors produced by gut-resident cells [8]. A previous study by Hwang et al. [45] demonstrated that IAP inhibited LPS-induced proinflammatory cytokine TNF-α and IL-6 production. We found that the treatment with IAP preferentially inhibited the plasma levels of proinflammatory cytokines TNF-α and IL-6 compared to control colitis mice fed with diet-induced obesity, indicating the potent anti-inflammatory potential of this enzyme especially in obese mice [20]. Moreover, this beneficial intervention attenuated plasma leptin levels. This hormone released mainly by adipose cells and enterocytes in the small intestine has been proposed to regulate energy balance by inhibiting hunger and diminishing fat storage in adipocytes. Leptin is also considered as pro-inflammatory cytokine linked with the impaired healing of TNBS-induced rats and mice fed a HFD [46,47,48,49]. Our study remains in keeping with these findings since we noticed a prominent increase in plasma leptin levels in obese mice with colitis, but this effect was reversed in exercising mice concomitantly treated with IAP. We propose herein that this combination of physical training on SW and IAP administration could be considered the most efficient intervention in the mechanism of abrogation of severity of colitis, mainly due to inhibition of plasma inflammatory markers such as MCP-1 [20]. Biologically active IL-12 is a heterodimeric cytokine composed of 2 subunits, p35 and p40, and is also known as IL-12p70 [50]. This cytokine is mainly produced by activated monocytes and macrophages and through activation of signal transducers that modulate the differentiation and maintenance of Th1 effector cells, the mechanism that leads to overexpression and release of IFN-γ, TNF-α and IL-17a [8]. These cytokines, in turn, promote T-cell responses and macrophage activation, thus contributing to enhanced inflammatory responses. Another cytokine, IL-17a, was reported to stimulate the epithelial and endothelial cells, fibroblasts and macrophages to produce other cytokines, e.g., IL-6, and to enhance antimicrobial peptide secretion and the expression of intestinal epithelial cells tight junction proteins. We found that both L-12 and IL-17a were most abundantly represented in obese mice with colitis, an effect that was clearly reduced in exercising mice with concomitant IAP administration. 

Besides barrier-enhancing functions, the anti-inflammatory cytokines mediate important immune-regulatory properties controlling immune system and microbial interactions. Mononuclear phagocytes play a role in integrating microbial cues to promote a regulatory T cells response controlled by IL-10 [8]. In our study, the highest plasma level of IL-10 was found in obese mice with intestinal inflammation. This can be interpreted as IL-10 induced promotion of an anti-inflammatory response in colitis mucosa confounded with obesity, while the combination of SW and IAP treatment affording intestinal protection resulted in a downregulation of this cytokine in intestinal mucosa. Moreover, another pleiotropic cytokine, IL-2, has been reported to contribute to immune alterations during inflammation and obesity [51,52]. However, even though the plasma level of IL-2 in our study was significantly lower in the HFD group, it only tended to increase in exercising mice treated with IAP as compared to colitis animals fed a standard diet. 

Oxidative stress has been shown to play an important role in the pathogenesis of damage to the intestinal mucosa [53], and the exacerbation of colitis by the implementation of high-fat diet has been observed [54]. The supply of dietary fat induces an increase in the intensity of lipid peroxidation in adipocyte and non-adipocyte cells [55], thereby increasing the oxidative stress of adipose tissue. Indeed, we observed a marked increase of the products of cell membrane lipid peroxidation in the colonic mucosa of the animals fed with an obesity-inducing diet compared to the corresponding groups fed the normal diet. The level of MDA and 4-HNE was clearly diminished in exercising obese mice with colitis with or without treatment with IAP, confirming that exercise alone or combined with IAP treatment potently reduced lipid peroxidation. Intensity of oxidative processes can be measured by the level of reduced glutathione, which is a key component of the endogenous antioxidant defense system in the intestine. The increase in GSH level stimulates the mobilization of internal systems of scavenging reactive oxygen species against noxious stimuli. Herein, the lowest content of GSH and GSSG in obese mice with or without colitis well corresponded with previous studies involving experimental colitis in rodents [56,57,58]. Interestingly, mice supplemented with IAP showed an increase in GSH and GSSG content, suggesting that the restoration of reduced glutathione could account for the beneficial effect of this enzyme. Furthermore, we observed that the activity of antioxidizing enzyme SOD was significantly higher in obese mice with colitis, but the combination of IAP and SW had diminished SOD activity. Long-standing inflammation causing oxidative stress is the most important risk factor for cancer development and therefore, 8-OHdG formation is considered as the most common indicator of DNA damage caused by oxidative stress being also associated with IBD and IBD inflammation leading to cancer development [59]. We found that the physical activity and IAP administration decreased the content of this marker compared to sedentary obese mice with colitis, indicating that this combination is superior among each of them applied alone against the development of DNA damage, thereby improving the healing of colitis in our study. 

In recent years, due to a development of DNA sequencing methods, with special emphasis on next generation sequencing techniques, more often a link has been reported between the host’s health conditions and the bacterial composition in a gut. It has been proved that total microbiome weight in humans reaches up to 2 kg and the number of bacterial cells in intestines 10-fold exceeds the number of host cells in humans [60]. The microbiome is defined as all the bacteria, viruses, fungi, archaea and eukaryotes that inhabit the host’s organism and the number of species is estimated at 5000 species [61]. The microbiome is considered a separate “organ” related to the metabolism and immune system. Gut microbiome encodes approximately 3 million genes responsible for the metabolism [62]. This relationship between imbalance in a microbiome composition and increased risk of certain diseases’ appearance, such as obesity, metabolic abnormalities and/or autoimmune diseases, has been emphasized. Furthermore, such a shift in bacterial composition may also be related to depression or cardiovascular disease [63,64]. In turn, introduction of increasingly popular fecal microbiota transplant from healthy individuals to the IBD-suffering individual offers a very promising therapeutic approach, among others, in colitis [65,66,67,68].

Our study demonstrated the domination of two phyla in all studied groups, namely, *Bacteroidetes* and *Firmicutes*. Similar results were reported by Laukens et al. and Turnbaugh et al. [69,70], who identified *Bacteroidetes* at 20–40% studied sequences and 60–80% sequences for *Firmicutes* in mice. In our studies, distribution of percentage varied in dependence of the studied group; nevertheless, total percentage was comparable. *Bacteroidetes* in the gut microbiome is directly related to lean body mass [71]. The relative proportion of *Bacteroidetes* increases proportionally with the body weight of obese animals. It has been reported that in rats and humans, an increase in the relative proportion of *Bacteroidetes* occur as a response to a HFD [72], a notion that has been confirmed in our study. The bacteria that represent *Proteobacteria* contain a gram-negative cell wall and thus present a primary endotoxin, LPS, to gut epithelial cells and immune cells. Compared to sedentary obese colitis animals, groups of exercising obese mice with or without IAP treatment have shown significantly lower relative proportion of *Proteobacteria*, suggesting the protective effect of voluntary exercise and IAP administration, possibly via inhibition of LPS and other endotoxins.

Analysis at the genus level showed the most divergent microbial composition in mice kept on a normal diet. Our data corresponded with observations described by Yun et al., which were carried out on children with various body mass indexes [73]. Introduction of a high fat diet effected an increase of the percentage of *Firmicutes* of 5% in comparison with an ND group, which is in keeping with evidence provided by Ley et al. [74]. In the literature, as a most abundant genus in mice microbiome, *Alistipes* is described [75]. We identified this genus in each group studied except obese mice with colitis, indicating the diminishment of *Alistipes* in microbiota composition in mice with colitis associated with obesity. It has been reported that *Clostridium* is related to IBDs, irritable bowel syndrome (IBS), colorectal cancer, allergies, neurological diseases and metabolic diseases [76]. We have observed an increased percentage of this genus in each group of mice except mice without colitis kept on ND or HFD. Interestingly, we reported the relative abundance of *Faecalibacterium* kept on a standard diet, which is known for its anti-inflammatory properties [77]. This might be due to the fact that obesity followed by a high fat diet may result in the development of an inflammation [78,79]. Surprisingly, no increase in percentage of *Lactobacillus* was detected in obese mice with access to moderate exercise combined with IAP treatment, and in turn, the highest percentage of this genus was noticed in obese mice with TNBS colitis. We hypothesize that this effect may be related to the acidophilic character of the *Lactobacillus* genus; colitis induction can create a favorable environment for these probiotic bacteria biofilms. Interestingly, *Ruminococcus*, which is implicated in Crohn’s disease, appeared as the most numerous in obese mice with colitis [80]. In contrast, we found that *Ruminococcus,* belonging to the *Ruminococcus* genera, occurred in a smaller percentage in colitis mice in comparison to healthy mice, but this effect was reversed in obese exercising mice with colitis treated with IAP. These results are consistent with results of rodent studies and human fecal transplant experiments in which *Ruminococcus* has been found to exert anti-inflammatory properties, restoring and maintaining normal gastrointestinal tract function and integrity [81,82]. For instance, *Ruminococcus gnavus* is a mucolytic bacteria found to be reduced in both UC and CD patients compared with normal subjects [83]. Nevertheless, the evidence-based medicine on this topic seems to be contradictory and more research should be undertaken to assess the real impact of *Ruminococcus* on the development of experimental and human IBD. 

Moderate exercise training-induced modification of the mice gut microbiota has been little studied before [84,85] and that is why we attempted to screen microbiota of obese mice with colitis with or without voluntary exercise and IAP administration. Evans et al. [71] and Allen et al. [84] have shown the reduction of *Turicibacter* confounded by obesity and voluntary exercise, and our data are in keeping with these results since the presence of *Turicibacter* was detected only in sedentary mice fed a standard diet. The reduction of *Turicibacter* microbial populations by voluntary exercise and HFD shown by few independent studies, still requires confirmation whether or not this particular bacterial population may be an emerging issue for obesity and physical training in human IBD [86,87]. The increase in the proinflammatory cytokines observed in obese colitis mice as well as changes in the microbiota can be due to an activation of the NF-κB transcription factor, the key pathway involved in intestinal inflammation of human CD [86]. Our findings are corroborative with a recent observation in the same TNBS model of animal colitis that SCFA-producing bacterial taxa such as *Ruminococaceae* and *Lachnospiraceae* are indeed reduced. Interestingly, the intragastric treatment with the synbiotic Syngut to attenuate the severity of experimental colitis counteracted this effect on SCFA-producing bacterial taxa [87].

In conclusion, the present study demonstrates that administration of IAP combined with moderate physical activity as an alternative intervention to pharmacotherapy significantly reduced gross and microscopic inflammatory response and oxidative stress markers in obese mice with colitis. The beneficial effect of this combination might be due to a decrease of expression and release of proinflammatory cytokines and attenuation of oxidative stress and DNA damage as well as changes in gut microbiome. It seems reasonable to investigate the abundance of *Ruminococcus* gut bacterial species and its therapeutic efficacies in the development of colitis when administered with IAP. The results from this study further support the notion on a potential role of IAP in strengthening of the protective effects caused by moderate physical activity against the inflammatory colonic damage in the animal model of colitis. Moreover, this study may also provide new insights into the mechanisms by which the combination of IAP and voluntary exercise can exert a beneficial effect on colitis under conditions of obesity, which definitely deserves future investigation in clinical scenario of IBD.

## 4. Materials and Methods

### 4.1. Animals and Diets

Animal studies were conducted on 45 female C57BL/6J mice, 10 weeks of age with an initial weight 18–22 g, kept in pathogen-free cages with constant temperature and control of ventilation and humidity (Bioscape Bio. A.S., Delft, The Netherlands) with free access to water and food and kept in laboratory conditions with 14 h/10 h day/night cycles. The study was approved by the local Ethical Committee at the Jagiellonian University Medical College in Cracow, Poland (No. 19/2016), and was run in accordance with the Helsinki declaration and with implications for replacement, refinement or reduction (the 3Rs) principle (Decision No.: 19/2016; date: 20 July 2016).

### 4.2. Experimental Design

Mice were subjected to an adaptation period for 3 days after the purchase, and after that, they were randomly assigned into 9 experimental series, each consisting of 6–8 animals per group: (1) sedentary mice kept on a normal diet (ND); (2) sedentary mice fed a high-fat diet (HFD); (3) mice fed a HFD and subjected to voluntary physical activity on a spinning wheel (HFD + SW); (4) mice fed a HFD with intestinal alkaline phosphatase (IAP) administration (HFD + IAP); (5) mice fed a HFD, subjected to voluntary physical activity on SW and administered with IAP (HFD + SW + IAP); (6) mice fed a HFD and with colitis induced by 2,4,6-trinitrobenzenesulfonic acid (TNBS) (HFD + TNBS); (7) mice with TNBS colitis fed a HFD and subjected to voluntary physical activity on SW (HFD + SW + TNBS); (8) mice with TNBS colitis fed a HFD with IAP administration (HFD + IAP + TNBS); and (9) mice with TNBS colitis fed a HFD, subjected to voluntary physical activity on SW and administered with IAP (HFD + SW + IAP + TNBS) (Figure 1). Control group (1) was fed regular chow pellets as a normal diet (ND, diet C 1000; Altromin, Lage, Germany). In case of groups 2–9, mice were fed for 14 weeks with a high-fat diet (HFD) (C 1090-70—obesity-inducing diet with 70% energy from fat (42% pork fat)) as described previously [3,20] and continued in our present study up to the end of experimentation at day 5 upon placebo or 2,4,6-trinitrobenzene sulfonic acid (TNBS) administration with the methods for colitis described below. After initial feeding for 14 weeks, animals fed with HFD from groups 3 (SW alone), 5 (SW +IAP), 7 (SW + TNBS) and 9 (SW + IAP) were subjected to the voluntary physical activity on a spinning wheel (no. 76-0413, Panlab, Harvard Apparatus, Holliston, MA, USA) for 7 weeks, and all mice were kept on ND (group 1) or HFD (groups 2–9) to the end of the experiment according to the timeline of our experimental design, presented in Figure 1. The mice had free access to the spinning wheel and were placed in individual cages—each connected to one wheel. Each wheel was attached to a device that counts the daily activity of mice (no. 76-0243, Panlab, Harvard Apparatus, Holliston, MA, USA). Then, mice from group 4 (HFD + IAP), group 5 (SW + IAP), group 8 (HFD + IAP + TNBS) and group 9 (SW + IAP + TNBS) were administered by IAP in a daily dose of 200 U (P0114, Sigma Aldrich, St. Louis, MO, USA), for 2 weeks, in drinking water. All IAP-treated animals still had the access to spinning wheels. During the time of experimentation, both the sedentary and exercising mice were still maintained on ND and HFD, respectively (Figure 1). After the time of 2 weeks of IAP administration, the experimental colitis was induced in all mice to assess the effect of a voluntary physical activity (SW) combined or not with IAP on the course of it. At day 5 upon colitis induction, animals of all groups were weighted, their grips strength was measured and their stool was collected for fecal microbiota assessment (Figure 14). For clarity and greater uniformity of our work, we present data from all 9 groups collected at the end of the experiment, that is, day 5 after rectal administration of TNBS or placebo (TNBS (−) control) (Figure 14). 

### 4.3. Grip Strength Test

Grip strength was measured using a grip strength test device (BIO-GS3, Pinellas Park, FL, USA). The device was placed on a heavy, metal base, several centimeters high to minimize external forces that could positively distort the results. Animals were placed in the center of a metal mesh to they could catch it with all four limbs. The mesh was placed outside the laboratory table to induce fear in mice from falling from a height and thus, to force the mouse to grip firmly the mesh. Then, each mouse was pulled horizontally to the ground from a metal element of the device so it measured the force needed to pull the mouse away and thus, how much grip strength the animal exhibited. The measurement was carried out three times, with a few minutes’ interval for each mouse. The arithmetic mean of the three measured values was determined as the correct measurement result. The unit of measurement was a newton [N]. Relative grip strength is expressed as mean grip strength for each group divided per mean body weight for each group [N/kg] [88].

### 4.4. Experimental Colitis Induction

Experimental colitis was induced in groups 6–9 by intra-colonic administration of TNBS as described before [20]. Briefly, a dose of 4 mg of TNBS (aqueous solution, no. 92822, Sigma Aldrich, St. Louis, MO, USA) was dissolved in 50% ethanol solution and applied rectally in a volume of 175 μL per mouse, using a soft polyethylene catheter (Instech, Plymouth Meeting, PA, USA) to a depth of about 4 cm. Animals were anesthetized by inhalation of isoflurane (2–3% in a breathing mixture with oxygen; Aerrane, PGF, Wroclaw, Poland) using the Ugo Basile compact anesthesiological system (No. 21100, Italy). Until awakening, the animals were placed in the Trendelenburg position to prevent the loss of TNBS solution. Animals from the control group received rectally analogous to TNBS volume of 50% ethanol with water in an amount corresponding to the aqueous solution of TNBS for a dose of 4 mg (approx. 14 µL) (placebo). All animals were given subcutaneous approximately 1 mL of saline to prevent dehydration.

At day 5 after colitis induction or placebo treatment, the animals were sacrificed with an i.p. lethal dose of pentobarbital (Biowet, Pulawy, Poland). The abdominal cavity was opened and the colon was separated. The disease activity index (DAI) was calculated using a modification of a previously published compounded clinical score. In brief, the DAI comprised the scoring for diarrhea and lethargy (0–3) and rectal bleeding assessment involved a visual inspection of blood in feces and the perianal area (0–5), wherein 0 represents a healthy colon and 5 represents the most intensive course of the disease, bleeding rectal and extensive ulcers [20]. Colonic tissue samples were collected on ice, snap-frozen in liquid nitrogen and stored at −80 °C until further analysis. Blood was drawn from the *vena cava* and the plasma concentrations of interleukin (IL)-2, IL-6, IL-10, IL-12p70, IL-17a, tumor necrosis factor-alpha (TNF-α), monocyte chemoattractant protein-1 (MCP-1) and leptin were assessed. About 200 mg of fecal samples were collected for each group of mice at day 5 of termination of TNBS colitis as well as in those given placebo for TNBS, and stored at −80 °C until further analysis.

### 4.5. Macroscopic and Microscopic Changes in the Colonic Mucosa Assessment

For histology determination, the colonic tissue was excised and fixed in 10% buffered formalin with pH = 7.4. Samples were dehydrated by passing them through a series of alcohols with incremental concentrations, equilibrated in xylene for 10–15 min and embedded in paraffin. Paraffin blocks were cut into about 4 μm sections using a microtome. The prepared specimens were stained with hematoxylin/eosin (H&E) and evaluated using a light microscope (AxioVert A1, Carl Zeiss, Oberkochen, Germany). Digital documentation of histological slides was obtained using the mentioned microscope equipped with an automatic scanning table and ZEN Pro 2.3 software (Carl Zeiss, Oberkochen, Germany). For histology assessment, formalin-fixed full-thickness samples were serially cross-sectioned (10 μm thick). Sections were scored using the histopathological score for intestinal inflammation in mice proposed by Erben et al. [89] 1: mild mucosal inflammatory cell infiltrates with intact epithelium, 2: inflammatory cell infiltrates into mucosa and submucosa with undamaged epithelium, 3: mucosal infiltrates with focal ulceration; 4: inflammatory cell infiltrates into mucosa and submucosa and focal ulceration, 5: moderate inflammatory cell infiltration into mucosa and submucosa with extensive ulcerations; 6: transmural inflammation and extensive ulceration.

### 4.6. Next Generation Sequencing of Gut Microbiome

#### 4.6.1. DNA Isolation

DNA was isolated from stool samples using QIAamp DNA Stool Mini Kits (Qiagen, Hilden, Germany) according to the manufacturer protocol. Approximately 100 mg of stool was mixed with 1 mL of InhibitEX buffer and homogenized by vortexing. The next samples were heated (95 °C for 5 min) and centrifuged. Aliquots of supernatant (200 µL) were transferred to fresh tubes containing Proteinase K (15 µL) and AL buffer (200 µL) and incubated at 70 °C for 10 min. As the final step, 200 µL of ethanol was added to each tube and DNA was recovered using QIAamp spin columns. Isolated DNA samples were next aliquoted and stored (−80 °C) for further analyses.

#### 4.6.2. 16s rRNA Sequencing and Data Analyses

Sequencing was performed with the use of an Ion Torrent Personal Genome Machine (PGM) platform (Life Technologies, Carlsbad, CA, USA) and 16S Metagenomics Kit (Life Technologies; Carlsbad, CA, USA) as described previously [90]. A maximum of 26 barcoded 16S samples were sequenced on an Ion 318 v2 chip (Life Technologies, Carlsbad, CA, USA) using the Ion PGM Sequencing 400 Kit (Life Technologies, Carlsbad, CA, USA) according to the manufacturer’s protocol.

Sequencing data were analyzed and bacterial taxonomy was evaluated with the use Ion Reporter software (Thermo Fischer, Waltham, MA, USA) and fallowing databases: Curated MicroSEQ(R) 16S Reference Library v2013.1; Curated Greengenes v13.5. Data are presented as a percentage of bacterial taxa in each sample. 

### 4.7. Lipid Peroxidation Determination

To measure the concentration of malondialdehyde (MDA) and 4-hydroxynonenal (4-HNE) in colonic samples, the spectrophotometric method using the kit for lipid peroxidation (Bioxytech, LPO-586, Oxis, Portland, OR, USA) was used [3]. The colonic mucosal sample weighing 94.11 ± 4.63 mg (mean ± SEM) was excised and transferred to a vial containing 10 μL of 0.5 M of butylated hydroxytoluene in acetonitrile—in order to prevent sample oxidation—and 1 mL of 20 mM PBS (pH = 7.4). Samples were subsequently mechanically homogenized for 15 s and the homogenates were centrifuged for 10 min (3000× *g* at 4 °C). The obtained clear supernatant was stored at −80 °C until assayed. Two independent biological replicates were performed. The colorimetric assay used to determine MDA concentration in colonic mucosa is based on the reaction of a chromogenic reagent (N-methyl-2-phenylindole) with MDA and 4-HNE at 45 °C, which yields a stable chromophore with maximal absorbance at 586 nm. The absorbance was analyzed with a microplate reader (Tecan Sunrise, Männedorf, Switzerland). Results were expressed as nanomoles per gram of colonic tissue (nmol/g).

### 4.8. Total Glutathione (GSH + GSSG) Concentration Measurement

To determine the concentration of a reduced form of glutathione (GSH), a colorimetric assay using an enzymatic recycling method (Cat# 703002, Glutathione Assay Kit Cayman Chemical, Ann Arbor, MI, USA) was used. The method is based on an enzymatic recycling method using glutathione reductase, in which a chromophoric thione with a maximal absorbance wavelength at 410 nm is obtained. The colonic sample of 64 ± 3.85 mg (mean ± SEM) was collected and homogenized in 1 mL of 50 mM MES, pH 6–7, containing 1 mM EDTA. The homogenates were centrifuged for 15 min (10,000× *g* at 4 °C). The upper clear aqueous layer was collected and deproteinated by adding an equal volume of the MPA Reagent, mixing and centrifugation. The supernatant was collected and 50 μL of the TEAM Reagent was added per 1 mL of supernatant. Two independent biological replicates were performed. The level of total glutathione was measured with maximal absorbance at 410 nm by a microplate reader (Tecan Sunrise, Männedorf, Switzerland). Results were expressed as nanomoles per gram of tissue (nmol/g).

### 4.9. Superoxide Dismutase (SOD) Activity Determination

For the measurement of the SOD activity, a sample of colonic mucosa weighing 53.22 ± 3.76 mg (mean ± SEM) was collected and homogenized in cold 20 mM HEPES buffer (pH = 7.2) containing 210 mM mannitol, 70 mM sucrose and 1 mM EGTA per gram of colonic tissue and centrifuged for 5 min (1500× *g* at 4 °C). The supernatant was collected and immediately assayed, using the colorimetric assay for assessment of SOD activity (Cat# 706002, SOD Assay Kit, Cayman Chemical, Ann Arbor, MI, USA) [3]. This kit utilizes a tetrazolium salt for detection of superoxide radicals generated by xanthine oxidase and hypoxanthine. The amount of enzyme needed to exhibit 50% dismutation of the superoxide radical is defined as one unit of SOD. The absorbance was measured by a microplate reader (Tecan Sunrise, Männedorf, Switzerland) at 450 nm and the results were expressed as units per gram of colonic tissue (U/g).

### 4.10. Measurement of 8-Hydroxy-2′-deoxyguanosine (8-OHdG) Concentration

For measurement of 8-OHdG content as DNA oxidation marker, the DNA was isolated from colonic mucosa using the ELISA kit (589320, Cayman Chemical, Ann Arbor, MI, USA) according to the manufacturer’s protocol [91]. Briefly, the collected tissue was immediately frozen in a liquid nitrogen and stored at −80 °C until analyzed. The collected colon fragments were homogenized in 0.1 M phosphate buffer (pH = 7.4) with the addition of 1 mM EDTA in the amount of 5 mL per 1 g of tissue. The samples were then centrifuged for 10 min at 1000× *g* and the supernatant was purified using the Tissue DNA Purification Kit, cat. No. E355, from EURx, Gdansk, Poland. Total DNA concentration was measured using a Qubit 3 fluorimeter (Thermo Fisher Scientific, Waltham, MA, USA). The obtained DNA was incubated with P1 nuclease and then adjusted to pH 7.5–8.5 with 1 M Tris. 1 unit of alkaline phosphatase per 100 µg DNA added and incubated for 30 min at 37 °C. The samples were then incubated at 100 °C for 10 min and placed on ice. The test sample, the 8-OH-dG-acetylcholinesterase conjugate and the monoclonal antibody were added to the plate coated with the goat polyclonal antibody. The plate was then sealed with a membrane and incubated at 4 °C for 18 h. After this time, the membrane was detached and Ellman’s reagent containing the substrate for acetylcholinesterase was added. The plate was resealed with the membrane and incubated for 120 min on an orbital shaker in the dark. The absorbance was measured with reagent blank and different concentration of standards with a microplate reader (Tecan Sunrise, Männedorf, Switzerland) at a wavelength of 412 nm. The 8-OHdG concentration was expressed as ng of 8-OHdG per 1 µg of total DNA.

### 4.11. Luminex Microbeads Fluorescent Assays

Determination of plasma IL-2, IL-6, IL-10, IL-12p70, IL-17a, TNF-α, MCP-1 and leptin levels was performed using Luminex microbeads fluorescent assays (Bio-Plex Pro™ Mouse Cytokine) and Luminex 200 system (Luminex Corp., Austin, TX, USA). Results were calculated from calibration curves and expressed in pg/mL of plasma blood for IL-2, IL-6, IL-10, IL-12p70, IL-17a, TNF-α, MCP-1 and in ng/of plasma blood for leptin, as described in detail previously [20].

### 4.12. Statistical Analysis

Results are expressed as means ± SEM. The data were processed by the GraphPad Prism 5.0 software (GraphPad Software Inc., La Jolla, CA, USA). Statistical analysis was conducted using Student’s *t*-test or ANOVA with Dunnett’s multiple comparison or Tukey’s post hoc test if more than two experimental groups were compared. The size of each experimental group was *n* = 6–8. Type I statistical error *p* < 0.05 was considered significant. Statistical analysis for microbiome data was performed with the use of the group_significance.py program implemented in Qiime software [92]. The goodness of fit g-test was calculated, which compares the ratio of the OTU frequencies in the sample groups.

## Figures and Tables

**Figure 1 ijms-23-02964-f001:**
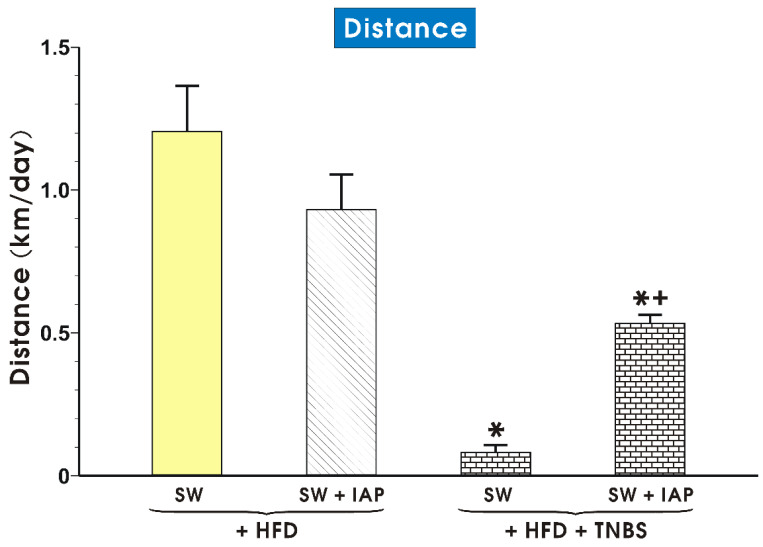
The running distance in obese exercising mice subjected to voluntary exercise on spinning wheels (SW) alone, SW combined or not combined with intestinal alkaline phosphatase (IAP) administration with or without TNBS colitis. Results are mean ± S.E.M. of 6–8 animals. An asterisk indicates a significant change (*p* < 0.05) as compared to the respective values in exercising mice fed a high fat diet (HFD). An asterisk and cross indicate a significant change (*p* < 0.05) as compared to the values in exercising obese mice with or without colitis.

**Figure 2 ijms-23-02964-f002:**
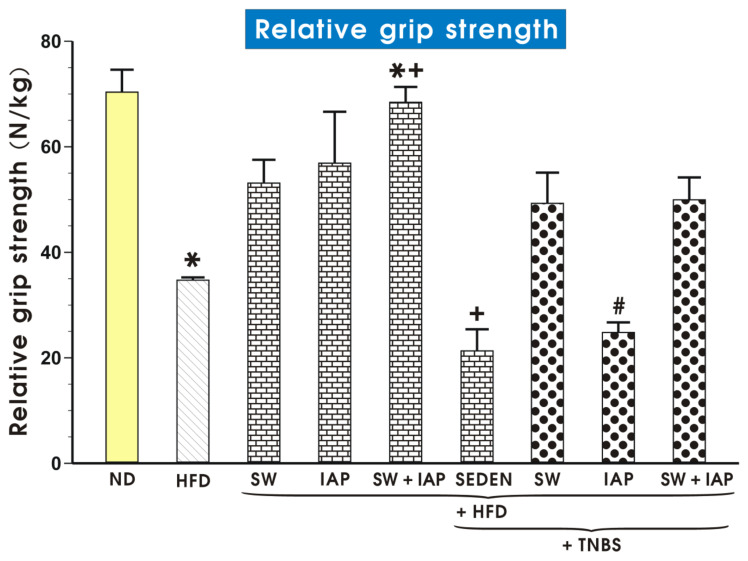
Relative grip strength in sedentary mice fed a normal diet (ND) or high fat diet (HFD) with or without TNBS colitis subjected to voluntary exercise on the spinning wheels (SW) or a treatment with the intestinal alkaline phosphatase (IAP) alone or combined with SW (SW + IAP). Results are mean ± S.E.M. of 7 animals per group. An asterisk indicates a significant change (*p* < 0.05) as compared to the respective values in mice fed ND. An asterisk with cross indicates a significant change (*p* < 0.05) as compared to the respective values in obese mice subjected to SW or treated with IAP alone. A cross indicates a significant change (*p* < 0.05) as compared to the respective values in HFD fed sedentary (SEDEN) mice. A hash indicates a significant decrease (*p* < 0.05) as compared to the respective values obtained in obese TNBS mice subjected to SW only or the combination of SW + IAP.

**Figure 3 ijms-23-02964-f003:**
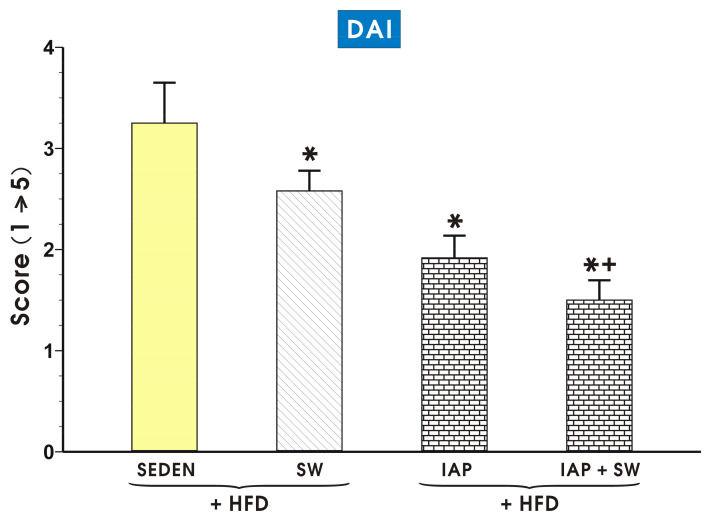
The effect of exercise on spinning wheels (SW) and intestinal alkaline phosphatase (IAP) administration both applied alone or in the combination SW+ IAP on the disease activity index (DAI) in obese mice (HFD fed) with TNBS-induced colitis as compared to sedentary (SEDEN) obese mice with TNBS colitis. Results are mean ± S.E.M. of 8 animals per each group. An asterisk indicates a significant change (*p* < 0.05) as compared to the respective value in the obese SEDEN mice with colitis. An asterisk and cross indicate a significant change (*p* < 0.05) as compared to the respective values in the obese mice with TNBS-induced colitis subjected to SW only or to the treatment with IAP alone.

**Figure 4 ijms-23-02964-f004:**
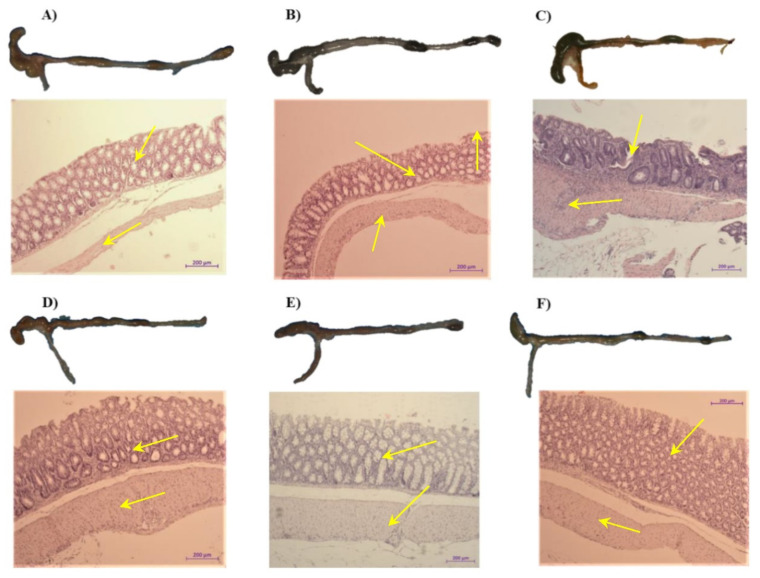
The representative macroscopic and microscopic appearance of the colon obtained from: (**A**) a sedentary mouse without colitis and fed a normal diet (ND); (**B**) a sedentary mouse without colitis and fed a high fat diet (HFD); (**C**) an obese mouse fed HFD with TNBS-induced colitis; (**D**) an obese mouse with TNBS colitis, subjected to exercise on spinning wheels (SW); (**E**) an obese mouse with the intestinal alkaline phosphatase (IAP) administration with TNBS colitis; (**F**) an obese mouse with TNBS colitis and subjected to the combination of exercise on SW and administered with IAP. The histology photomicrographs represent: (**A**) a colon from a sedentary mouse without colitis and fed a ND: normal macroscopic appearance of the colonic mucosa and no histologic injury to mucosa and submucosa as indicated by arrows; (**B**) a colon from a sedentary mouse fed a HFD: normal macroscopic appearance of the colonic mucosa and no histologic injury to mucosa and submucosa as indicated by arrows, (**C**) a colon from an obese, sedentary mouse with colitis: intestinal damage, hemorrhagic lesions, prominent mucosal inflammation and a bloody effusion of colonic mucosa and submucosa (arrows); (**D**) a colon from an obese mouse with voluntary exercise on SW and TNBS-induced colitis: fewer hemorrhagic lesions and mild inflammatory reaction in submucosa (arrows); (**E**) a colon from an obese mouse with IAP administration and TNBS-induced colitis: less inflammatory reaction in mucosa and submucosa, signs of regeneration of hypertrophic mucosa (arrows); (**F**) a colon from an obese mouse subjected to the combination of exercise on SW and IAP treatment with TNBS-induced colitis: marked improvement in histopathology of colonic mucosa as manifested by a preservation of mucosal lining, hypertrophic epithelium and a reduction in bleeding and hemorrhagic lesions (arrows).

**Figure 5 ijms-23-02964-f005:**
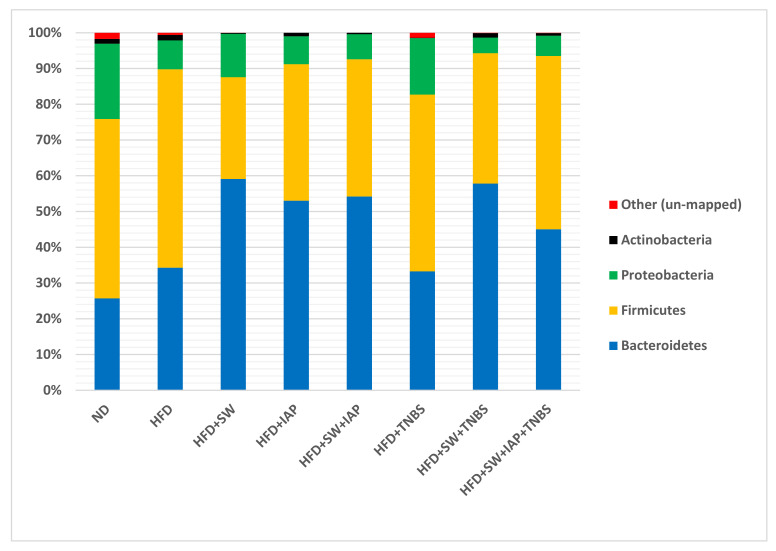
Relative abundance (%) of the gut microbiome at the phylum level for mice fed a normal diet (ND) only (control) or a high fat diet (HFD) with or without TNBS colitis administered with intestinal alkaline phosphatase (IAP) and subjected or not to voluntary exercise on spinning wheels (SW).

**Figure 6 ijms-23-02964-f006:**
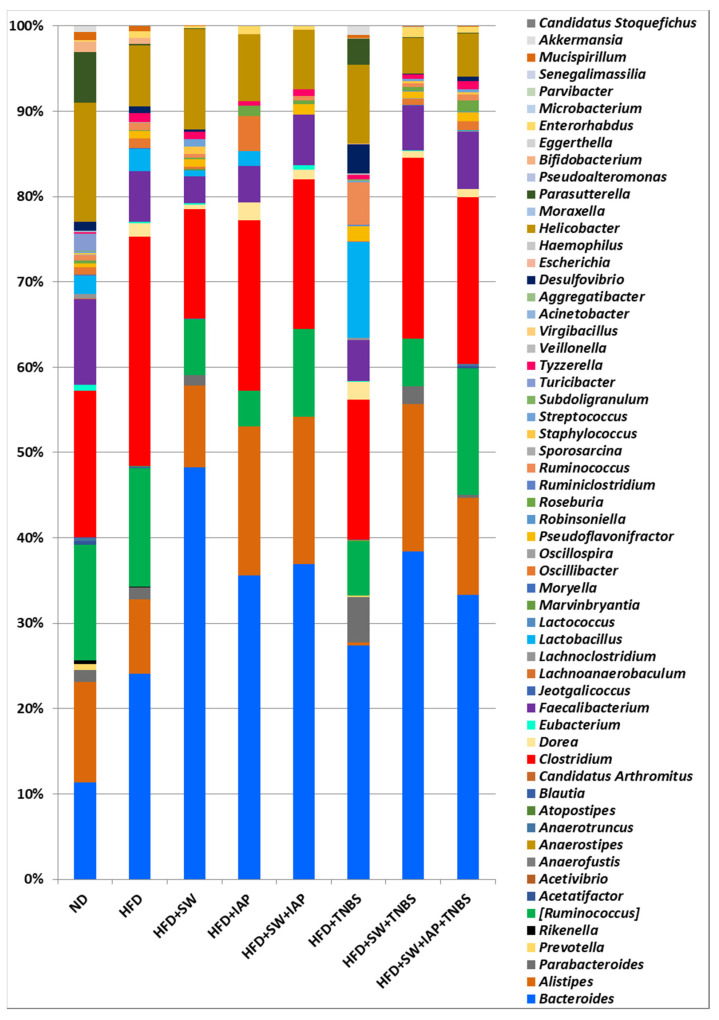
Relative abundance (%) of the gut microbiome at the genera level for mice fed a normal diet (ND) only or fed a high fat diet (HFD) with or without TNBS-induced colitis administered with intestinal alkaline phosphatase (IAP) alone, or exercising on spinning wheels (SW) only or subjected to the combination of SW and IAP (SW + IAP).

**Figure 7 ijms-23-02964-f007:**
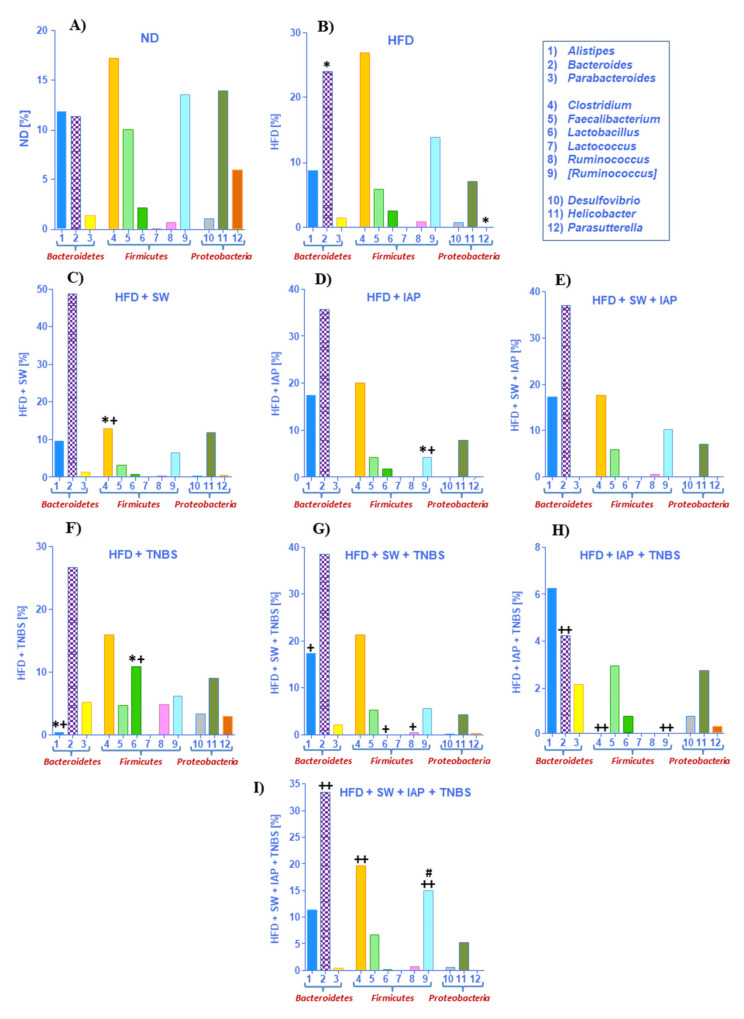
Relative abundance of the gut microbiome for the most important genus of mice fed normal diet (ND) (**A**) or high fat diet (HFD) (**B**) only, obese mice with spinning wheel (SW) exercise (HFD + SW) (**C**), obese mice administered with intestinal alkaline phosphatase (IAP) alone (HFD + IAP) (**D**) or subjected to the combination of SW and IAP without TNBS colitis (HFD + SW + IAP) (**E**) and obese mice with TNBS colitis (HFD + TNBS) (**F**), obese mice with TNBS colitis exercising on SW (HFD + SW + TNBS) (**G**), obese mice with TNBS colitis administered with IAP alone (HFD + IAP + TNBS) (**H**) or obese mice administered with IAP in combination with SW (HFD + SW + IAP + TNBS) (**I**). An asterisk indicates a significant change (*p* < 0.05) as compared to the respective values in the sedentary mice fed ND. An asterisk and cross indicate a significant change (*p* < 0.05) as compared to the respective values in the sedentary mice fed a HFD. A cross indicates a significant change (*p* < 0.05) as compared to the respective values in the HFD fed mice with TNBS colitis. A double cross indicates a significant change (*p* < 0.05) as compared to the respective values in HFD fed mice with or without access to SW with TNBS colitis. Hash indicates a significant change (*p* < 0.05) as compared to the respective values in HFD fed TNBS colitis mice with SW only or treatment with IAP alone.

**Figure 8 ijms-23-02964-f008:**
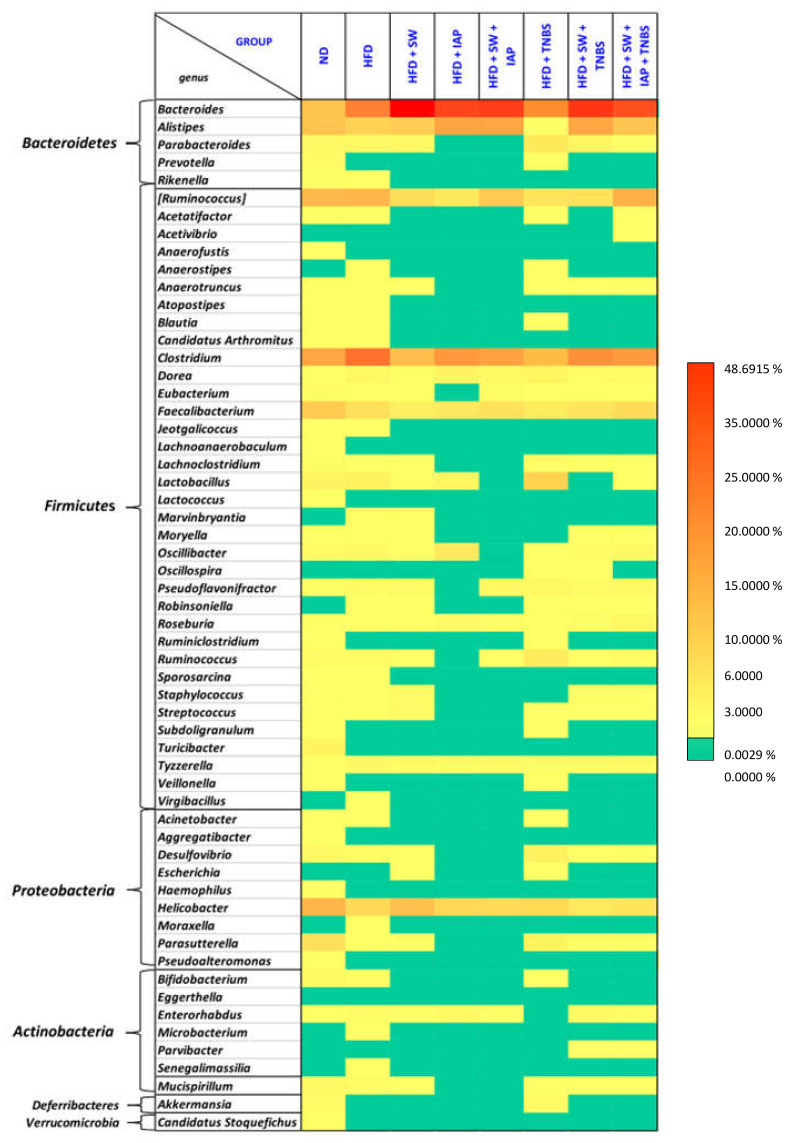
Heat map of species abundance clustering at the genus level, estimated from 16S rRNA amplicon sequencing, of gut microbiome of mice in mice fed a normal diet (ND) or high fat diet (HFD) alone, in mice fed a HFD and subjected to voluntary exercise on spinning wheels (SW) (HFD + SW), mice fed a HFD and treated with the intestinal alkaline phosphatase (IAP) (HFD + IAP), mice fed a HFD with TNBS colitis (HFD + TNBS), mice fed HFD with IAP administration (HFD + SW + IAP), mice fed HFD with access to SW with TNBS colitis (HFD + SW + TNBS) and in TNBS colitis mice fed HFD with the combination of SW and IAP administration (HFD + SW + IAP + TNBS). Color reflects relative abundance from low (yellow) to high (red) with green as zero.

**Figure 9 ijms-23-02964-f009:**
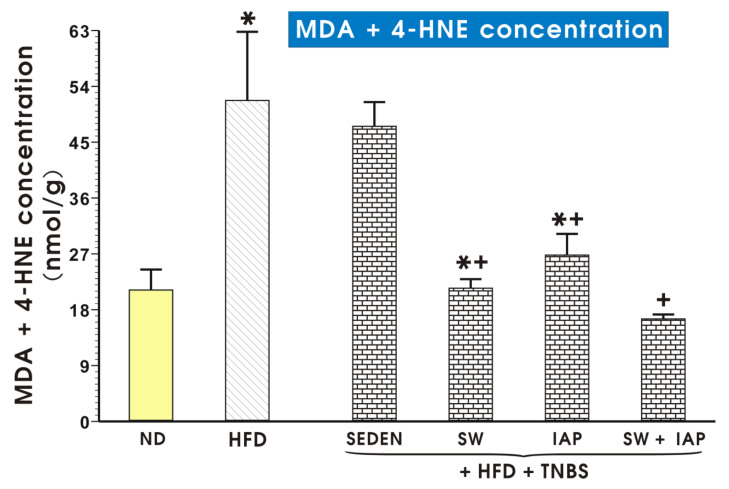
Lipid peroxidation products expressed as malondialdehyde and 4-hydroxynonenal (MDA + 4-HNE) concentration in colonic mucosa of a normal diet (ND) or obese mice, voluntary exercised TNBS colitis mice exercising on spinning wheels (SW) with or without the intestinal phosphatase (IAP) treatment or in those subjected to the combination of SW and IAP. Results are mean ± S.E.M. of 6–8 animals per group. An asterisk indicates a significant change (*p* < 0.05) as compared to the respective values in the sedentary mice (SEDEN) fed ND. An asterisk with cross indicates a significant change (*p* < 0.05) as compared to the respective values in the sedentary mice fed a HFD with induced colitis. A cross indicates a significant change as compared to the respective values in the HFD mice with administration of IAP and induced colitis (*p* < 0.05).

**Figure 10 ijms-23-02964-f010:**
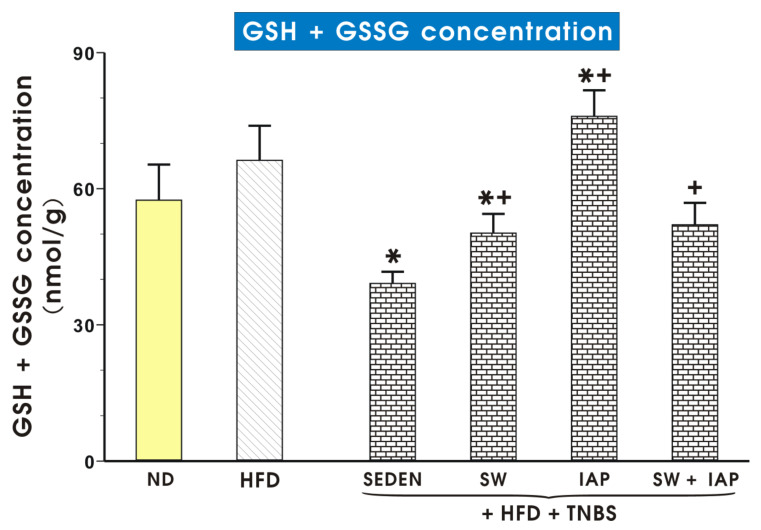
Total glutathione (reduced GSH+ oxidized GSSG) concentration in colonic mucosa in mice with or without TNBS colitis fed a normal diet (ND) or a high fat diet (HFD) with access to spinning wheels (SW), administered with or without combination with the intestinal alkaline phosphatase (IAP). Results are mean ± S.E.M. of 6–8 animals per each group. An asterisk indicates a significant change (*p* < 0.05) as compared to the respective values in the sedentary mice (SEDEN) fed HFD. An asterisk with cross indicates a significant change (*p* < 0.05) as compared to the respective values in the sedentary TNBS colitis mice fed a HFD. A cross indicates a significant change as compared to the respective values in obese TNBS mice treated with IAP (*p* < 0.05).

**Figure 11 ijms-23-02964-f011:**
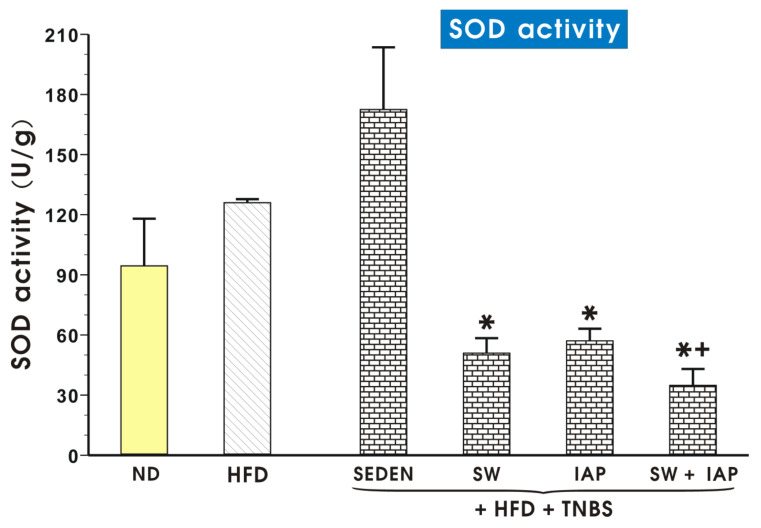
Determination superoxide dismutase (SOD) activity in colonic mucosa of a normal diet (ND) or obese mice fed a high fat diet (HFD) with or without TNBS colitis with access to spinning wheels (SW) with or without the intestinal alkaline phosphatase (IAP) treatment. Results are mean ± S.E.M. of 6–8 animals per group. An asterisk indicates a significant change (*p* < 0.05) as compared to the respective values in sedentary (SEDEN) mice fed a HFD. An asterisk with a cross indicates a significant change (*p* < 0.05) as compared to the respective values in the HFD fed mice with colitis subjected to SW only or treated with IAP alone.

**Figure 12 ijms-23-02964-f012:**
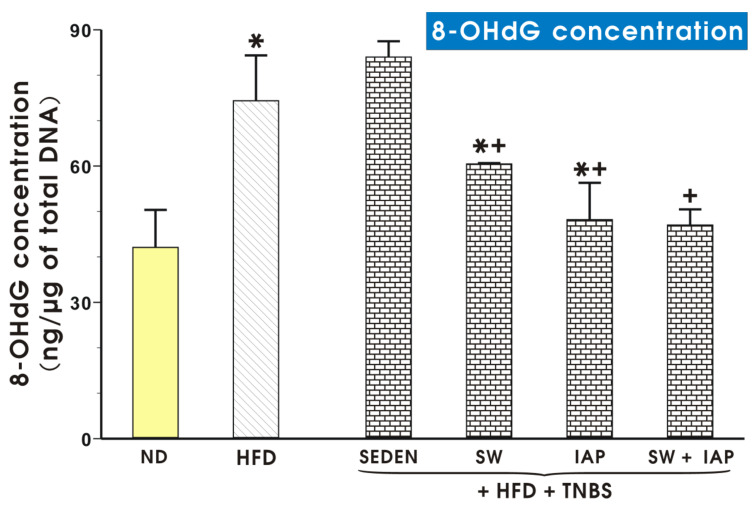
DNA oxidative damage in colonic mucosa of a normal diet (ND) or a high fat diet (HFD) mice with or without TNBS colitis subjected to exercise on spinning wheels (SW) or the intestinal alkaline phosphatase (IAP) treatment alone or the combination of SW and IAP. Results are mean ± S.E.M. of 6–8 animals per group. An asterisk indicates a significant change (*p* < 0.05) as compared to the respective values in the control mice fed an ND. An asterisk with cross indicates a significant change (*p* < 0.05) as compared to the respective values in the sedentary mice (SEDEN) with TNBS-induced colitis fed a HFD. A cross indicates a significant change (*p* < 0.05) as compared to the respective values in the HFD fed colitis mice with access to SW.

**Figure 13 ijms-23-02964-f013:**
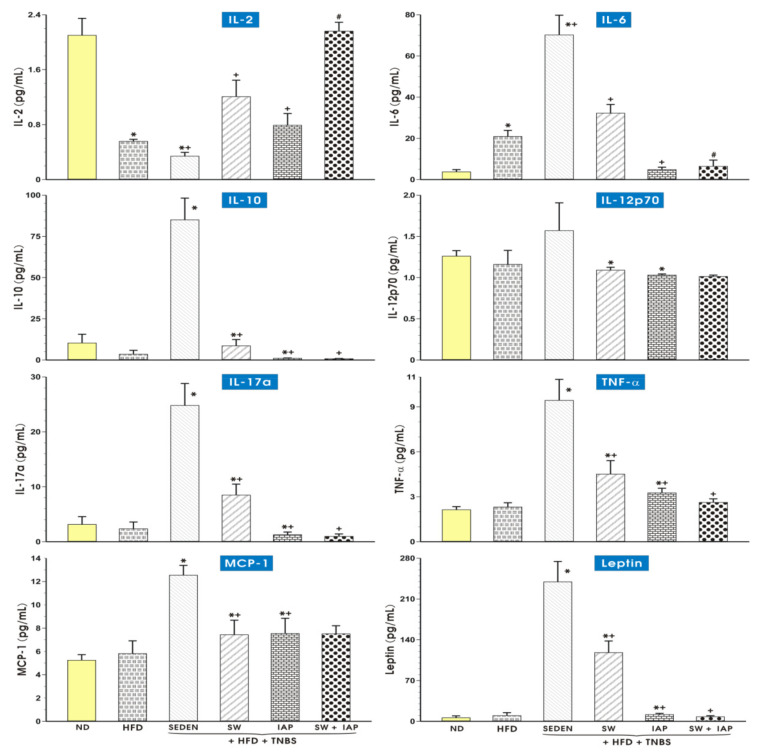
Plasma level of interleukin-2 (IL-2), IL-6, IL-10, IL-12p70, IL-17a, tumor necrosis factor-alpha (TNF-α), monocyte chemoattractant protein-1 (MCP-1) and leptin in mice fed a normal diet (ND) or a high fat diet (HFD) with experimental TNBS colitis exercising on spinning wheels (SW) and administered with IAP alone or combined with SW. Results are mean ± S.E.M. of 6–8 animals per group. An asterisk indicates a significant change as compared to the respective values in the sedentary mice fed ND for IL-2 and IL-6, as compared to the respective values in the sedentary mice fed HFD for IL-10, IL-17a, TNF-α, MCP-1 and leptin, and as compared to the respective values in the sedentary mice fed HFD with TNBS colitis for IL-12p70. An asterisk with a cross indicates a significant change (*p* < 0.05) as compared to the respective values in the sedentary mice fed a HFD for IL-2 and IL-6, as compared to the respective values in the sedentary mice fed HFD with TNBS colitis for IL-10, IL-17a, TNF-α, MCP-1 and leptin, and as compared to the respective values in the mice fed HFD access to SE and TNBS colitis for IL-12p70. A cross indicates a significant change (*p* < 0.05) as compared to the respective values in the sedentary mice fed HFD with TNBS colitis for IL-2 and IL-6, as compared to the respective values in the sedentary mice fed HFD with access to SW and TNBS colitis for IL-10, IL-17a, TNF-α and leptin. A hash indicates a significant change (*p* < 0.05) as compared to the respective values in the obese animals exercising on a spinning wheel with colitis for IL-2 and IL-6.

**Figure 14 ijms-23-02964-f014:**
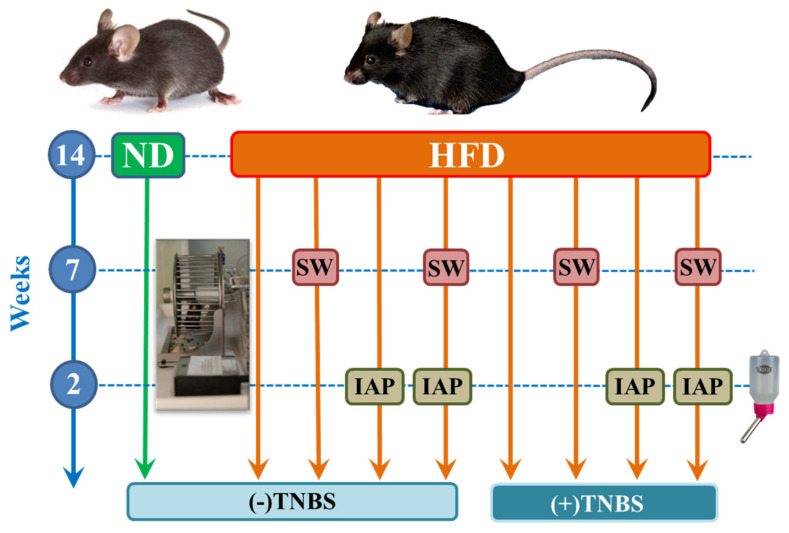
A time-line schedule presenting the experimental groups of mice fed a normal diet (ND) without any treatment and a high fat diet (HFD), exercising or not exercising on spinning wheels (SW), treated with intestinal alkaline phosphatase (IAP) alone or subjected to the combination of SW and IAP treatment, with or without 2,4,6-trinitrobenzene sulfonic acid (TNBS)-induced colitis.

**Table 1 ijms-23-02964-t001:** Body weight expressed in grams, in mice fed a normal diet (ND) alone or fed a high fat diet (HFD) alone, obese mice exercising on spinning wheels (SW) (HFD + SW), obese mice administered with intestinal alkaline phosphatase (IAP) alone and obese mice subjected to the combination of SW and IAP treatment (HFD + SW + IAP) with or without TNBS colitis. Results are mean ± S.E.M. of 6–8 animals per each group. An asterisk indicates a significant change (*p* < 0.05) compared to the respective values in the sedentary mice fed ND. An asterisk and cross indicate a significant change (*p* < 0.05) as compared to HFD fed mice. A cross indicates a significant change (*p* < 0.05) as compared to the respective values in HDF fed mice, obese TNBS sedentary mice or those exercising on SW only or treated with IAP alone or subjected to the combination of SW and IAP with TNBS colitis.

Group	Body Weight (g)
ND	25.53 ± 0.99
HFD	43.85 ± 1.68 *
HFD + SW	38.22 ± 2.09 *^+^
HFD + IAP	32.03 ± 6.28 *^+^
HFD + SW + IAP	31.20 ± 1.64 *^+^
HFD + TNBS	37.19 ± 2.68 *^+^
HFD + SW + TNBS	34.00 ± 4.28 *^+^
HFD + IAP + TNBS	44.81 ± 2.09 ^+^
HFD + SW + IAP + TNBS	35.13 ± 2.39 *^+^

## Data Availability

The data underlying this article will be shared on reasonable request to the corresponding author.

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
