# Peer review of "The Combination of Intestinal Alkaline Phosphatase Treatment with Moderate Physical Activity Alleviates the Severity of Experimental Colitis in Obese Mice via Modulation of Gut Microbiota, Attenuation of Proinflammatory Cytokines, Oxidative Stress Biomarkers and DNA Oxidative Damage in Colonic Mucosa"

_ijms, 2022, doi:10.3390/ijms23062964_

Round 1
Reviewer 1 Report
The authors have addressed all of my comments and concerns in the revised version.
Only a minor comment: the description of Fig. 8. in the text still does not match the panels of Fig. 8. (page 15 line 21: „HFD+TNBS (Figure 8E)”, lines 12-14 from bottom: „the combination SW and IAP (SW+IAP) without TNBS colitis (Figure 8F) or those with TNBS-induced colitis subjected to SW alone (Figure 8G) or the combination of SW+IAP (Figure 8H) (SW+IAP) without TNBS colitis (Figure 8F)”.
Author Response
Author’s reply to Reviewer Report (Reviewer 1)
We would like to thank Reviewer #1 for a minor but important critical comment, pointing to our inconsistency in the text as far as the description of Figure 8 is concerned.
We have now corrected this text error on page 15 by highlighting the red color change.
Reviewer 2 Report
Thank you very much for the revision. All my concerns/requests have been resolved by the author.
Author Response
Author’s reply to Reviewer Report (Reviewer 2)
We are grateful to Reviewer #2 for all comments related to the submission by us of a revised version of this manuscript, which had improved our MS in revised version and finally was now found satisfactory for this Reviewer.
Reviewer 3 Report
Dear authors,
Thank you for all your answers. They were mostly efficiently answered. Since you wrote that most of ND experiment will you not put in the manuscript, you can still add those data in supplementary section.
Author Response
Author’s reply to Reviewer Report (Reviewer 3)
We express our sincere gratitude to the Reviewer for her/his positive response to our revised version.
As suggested by this Reviewer, we have added the available data from ND-fed mice to the supplementary section of this manuscript. The results presented in the supplementary file just after the end of References are briefly described in the text of the Results and highlighted red.